METHODS

# Optimizing strategies for slowing the spread of invasive species

**Adam Lampert** *

Institute of Environmental Sciences, The Robert H. Smith Faculty of Agriculture, Food and Environment, The Hebrew University of Jerusalem, Rehovot, Israel

* adam.lampert@mail.huji.ac.il

**Data Availability Statement:** Data Availability: No external database is used in this paper. Simulation results as raw data can be accessed via Dryad: https://doi.org/10.5061/dryad.dfn2z356h Code Availability: The complete code can be accessed via Zenodo: https://doi.org/10.5281/zenodo.7683338.

## Abstract

Invasive species are spreading worldwide, causing damage to ecosystems, biodiversity, agriculture, and human health. A major question is, therefore, how to distribute treatment efforts cost-effectively across space and time to prevent or slow the spread of invasive species. However, finding optimal control strategies for the complex spatial-temporal dynamics of populations is complicated and requires novel methodologies. Here, we develop a novel algorithm that can be applied to various population models. The algorithm finds the optimal spatial distribution of treatment efforts and the optimal propagation speed of the target species. We apply the algorithm to examine how the results depend on the species' demography and response to the treatment method. In particular, we analyze (1) a generic model and (2) a detailed model for the management of the spongy moth in North America to slow its spread via mating disruption. We show that, when utilizing optimization approaches to contain invasive species, significant improvements can be made in terms of cost-efficiency. The methodology developed here offers a much-needed tool for further examination of optimal strategies for additional cases of interest.

## Author summary

In light of the global spread of invasive species that threaten ecosystems, biodiversity, agriculture, and human health, we developed an advanced computer algorithm to identify the optimal strategy to slow the spread of established invaders. In particular, the algorithm finds the most cost-effective way to allocate resources for treatment across different locations. The algorithm is generic and is suitable for a wide variety of population dynamical models and treatment methods. We tested the algorithm using both a broad-based model and a specific model focused on the spongy moth in North America. Our findings revealed significantly improved strategies for slowing the spread of invasive species. The algorithm thus offers a promising tool for improving environmental conservation and assisting policymakers in facing the challenges posed by invasive species.

**Funding:** The research was supported by the following funding sources to A.L.: Israel Science Foundation, grant no. 1180/23 (https://www.isf.org.il); and Ministry of Agriculture - Chief Scientist Office, Israel, grant no. 12-02-0048 (https://agriscience.co.il). The funders had no role in study design, data collection and analysis, decision to publish, or preparation of the manuscript.

**Competing interests:** The author has declared that no competing interests exist.

## Introduction

Due to global change, many species have become established outside their native habitats [1–3]. Some of these invasive species are harmful, inflicting severe damage to agriculture, forestry, biodiversity, human health, and infrastructure [1,2,4–7]. Commonly, such invasive species begin their infestation from a certain area where they become established and spread outward [8–10]. Biosecurity measures often combine prevention, detection, and early eradication to prevent the establishment of invasive species [11–13]. If these efforts fail and the invasive species becomes established, it often becomes practically impossible to entirely remove it, but it may still be possible to contain it to prevent or slow down its further spread to nearby areas [9,10,14,15]. Accordingly, containment projects are implemented worldwide to slow the propagation speed of various invasive populations. To name but a few examples, this includes populations of various insects, such as the spongy moth (*Lymantria dispar*) [15,16] and the emerald ash borer (*Agrilus planipennis*) in the US [16,17]; various marine species, such as the crayfish species *Procambarus clarkia* [18] and zebra mussels (*Dreissena polymorpha*) in the US and in Europe [19]; and numerous invasive plants worldwide [20].

In turn, containment often necessitates treatment over large areas and includes surveillance and eradication in regions close to the invasion area, suppression of the population in regions where it has already been established, and limiting the dispersal of the species into uninvaded areas at the borders between these regions to reduce propagule pressure, where all these treatment strategies are interrelated and can be used simultaneously [21]. A major question is, therefore, how managers should distribute the treatment efforts over space and time to minimize the net cost due to both treatment and damage. In particular, at what speed should the manager allow the species to propagate in order to minimize the net cost, and what is the optimal spatial allocation of the treatment along the front line where the species propagates? Researchers have integrated knowledge about population dynamics and economic costs into bioeconomic models of invasive species [10,22–24]; but to date, the primary focus of that literature has been on management in relatively small geographic areas. In such cases, the manager needs to either eradicate the population (remove it entirely from the target area) [25–29] or control it at a certain density [30–32]. Complete eradication is generally feasible only if the species' density is still low and the species is easily detectable even when its density is low [25,33], or if the species is subject to an Allee effect, namely, its population declines or grows more slowly when its density is low [28,29,34,35].

Several bioeconomic studies have explored the optimal containment of an invasive species to stop or slow its spread. Some of these studies have examined where to build fences to contain the species [36], some examined the optimal strategies for the surveillance of small populations of the species [37–39], and some examined the optimal distribution of treatment efforts to completely stop the spread of the species [21,40–43]. In turn, models that examined the optimal propagation speed have considered the cost of slowing the spread to a given speed, without an explicit model of the population's spatial dynamics and treatment [44].

One factor that limits studies that explore the optimal spatial distribution of treatments for slowing population spread is that standard optimization algorithms are limited in their capacity to solve spatially extended dynamical optimization problems over large areas. For example, Pontryagin's maximum principle [32,45] has been used to study certain spatially-extended bioeconomic models [41,46], but its application for solving more complex models requires its implementation specifically tailored to the specific model, and it might not be applicable in practice to all cases and boundary conditions, such as cases that incorporate dispersal kernel instead of diffusion. Also, methods like Dynamic Programming, which work well for finding the optimal treatment of local populations [47], often operate too slowly for solving models that incorporate large landscapes, and therefore, approximation algorithms might be necessary [48].

Here, we consider a population of an invasive species that spreads over space via some dispersal kernel, and we develop a novel algorithm that finds, at least approximately, (a) the optimal distribution of the treatment over space to slow the spread of the population to a certain speed and (b) the optimal target speed at which the species should be allowed to propagate. In particular, the target speed could be (a) zero (stopping the spread), (b) the natural propagation speed of the species (no treatment), (c) some other positive speed (slowing the spread), or (d) some negative speed (reversing the spread or eradicating).

The algorithm is general and enables the examination of a large variety of models. Here we demonstrate how the algorithm is used to conduct a comparative analysis, examining how the response of the species to the treatment method affects the optimal treatment strategy. A particular case study that we analyze is the management of the spongy moth (*Lymantria dispar*) to slow its population spread in North America [15]. In that case, we consider mating disruption as the main treatment method. In addition, we analyze a general and simpler model to further examine how the response of the species to the treatment affects the optimal treatment. One case that we examine using the general model is where the number of individuals abated per dollar invested does not depend on the species' density. This assumption is reasonable in models of large plants that are observable and thus easily detectable. It has been applied in various models of optimal eradication of *Spartina alterniflora* [25] and hybrid *Spartina* [27]. We also examine cases where the number of individuals abated per dollar invested decreases as the species density declines. This characterizes scenarios in which it is difficult to detect the species' individuals and an aerial treatment is necessary, such as using pesticides for treating insects [26,29,32].

## Model and methods

### Defining the problem

We study an invasive species population with a density $n(x,t)$ that can vary over a one-dimensional space axis, $x$, and time, $t$. The investment in treatment that suppresses the population, $A(x, t)$, can also vary over space and time. We assume that, initially, $n$ has the shape of a population "front," in which the population has reached its carrying capacity in a certain region (where the $x$ is large), and has not yet invaded regions that are further away from it ($n$ is close to zero where $x$ is small). Without any treatment ($A = 0$), the population propagates to the uninvaded area, such that the population front approaches a shape that propagates leftward at a certain speed, $v_0$. However, by applying treatment ($A > 0$), the manager can slow down the propagation speed to some $v < v_0$. The manager might even stop the speed entirely ($v = 0$), or reverse the direction of the front's propagation such that the invasive species would retreat ($v < 0$).

The goal of the algorithm developed here is to find the treatment function $A$ that slows the propagation speed down to a given, constant speed $v$, while using the minimal amount of annual investment. In turn, by using the algorithm to examine various values of $v$, we can obtain a comprehensive analysis of the best strategy, revealing the optimal target speed $v$ and the distribution of treatment over space to achieve that speed.

Before we describe the algorithm, we describe here the bioeconomic model, which comprises: (a) a model of the dynamics of the invasive species, including its response to the treatment; (b) an objective function, defining the goal of the manager (minimizing the total costs due to treatment and damage); and (c) the constraints—restrictions on the set of possible controls. The algorithm is general and could be applied to a wide variety of cases, and accordingly, we define here a general framework defining the dynamics of $n$. At the same time, we describe in detail how we parametrize the case study of the spongy moth management in North America and we demonstrate how the algorithm is applied to that case.

## Population dynamics

We consider a widely-used population model in which the population is spread over space and disperses according to a kernel function [49–51]. The underlying assumption is that, in addition to the adult population, the population may comprise propagules (juveniles, larvae, or seeds) that disperse over space; however, the development from propagules to adult is relatively fast, allowing us to consider a single variable, $n(x, t)$, characterizing the adult population density. We assume that treatments affect adults or propagules after their dispersal and we distinguish between (a) continuous-time dynamics, characterizing a population with overlapping generations, and (b) discrete-time dynamics, characterizing a population with no overlapping generations, such as that of univoltine insects or annual plants.

Specifically, if the dynamics are time-continuous, they are given by [21,49,50,52]

$$\frac{dn(x, t)}{dt} = -d(n) - R(n, A) + \int_{-\infty}^{\infty} b(n(x', t))G(x - x')dx', \tag{1}$$

where $d(n)$ is the rate of adult natural mortality, $R(n, A)$ is the rate of adult removal due to the treatment, $b(n)$ is the rate of propagule production, and $G$ is the dispersal kernel that determines how propagules disperse over space [51]. Note that the rates $d$, $R$, and $b$ vary over time and space due to their dependency on $n(x, t)$ and $A(x, t)$. In turn, if the dynamics are time-discrete, they are given by

$$n(x, t + 1) = \int_{-\infty}^{\infty} b(n(x', t), A(x', t))G(x - x')dx'. \tag{2}$$

This characterizes a population in which the new generation replaces the old one and dispersal occurs at some particular life stage, commonly as seed or larva, and $t$ is measured in units of generation, commonly a year. In the discrete-time scenario, we assume that $b$ is sufficiently small, such that the local dynamics converge to some steady state and do not exhibit oscillatory or chaotic dynamics.

## Functional forms–simple model

The method developed here for solving the model is general, but to demonstrate the results, we consider particular functional forms in the simulations. We begin here with a simple and general model, while in the next sub-section we describe the model that applies to the spongy moth. For the simple model, we consider continuous-time dynamics (Eq (1)) with the following birth and death rates [21]:

$$d = \gamma n, \tag{3A}$$

$$b = rn\left(1 - \frac{n}{k}\right), \tag{3B}$$

where $\gamma$ is the rate of adult natural mortality, $r$ is the per-capita rate of juvenile production when the population is small, and $k$ is the carrying capacity. We consider the following, widely-used form of the dispersal kernel function [51]:

$$G(x - x') = \frac{1}{\sqrt{2\pi}\sigma} \exp\left(\frac{(x - x')^2}{2\sigma^2}\right), \tag{4}$$

where $\sigma$ is the characteristic dispersal length.

In turn, the response of the population to the treatment, $R(n, A)$, is monotone increasing with $n$ and $A$ and satisfies $R(n, 0) = 0$. We consider the following, general form, that has been

used previously in the literature [41,53]:

$$R = \beta n^{\alpha} A,$$

(5)

where $\beta n^{\alpha}$ is the marginal efficiency of the treatment (number of individuals removed per dollar investment), and $0 \leq \alpha \leq 1$ is the species' invisibility, determining how the efficiency of the treatment diminishes as $n$ decreases. Specifically, $\alpha = 0$ characterizes cases in which the number of individuals abated per dollar invested does not depend on the species' density. This functional form has been applied in various models of optimal eradication of *Spartina alterniflora* [25] and hybrid *Spartina* [27]. These plants are easily detectable, and consequently, the cost of abating a given area covered by the species does not depend on how much of the total area has already been removed [25,27]. On the other hand, $\alpha > 0$ characterizes cases in which the number of individuals abated per dollar invested decreases as the species density declines [41,53]. In particular, $\alpha = 1$ characterizes cases in which the efficiency is proportional to $n$, which has been used to model treatment via pesticides that remove a certain fraction of the population in a given treatment [29,32].

### Functional forms–spongy moth population model

The spongy moth is a univoltine insect whose new generation replaces the previous one each year. Accordingly, its population can be characterized by the discrete-time dynamics, Eq (2). To slow the spread of the spongy moth in North America, mating disruption has emerged as a prevalent strategy [26,54]. Mating disruption involves the distribution of synthetic pheromone flakes throughout the affected zones during the mating season. The flakes mimic the female moth's natural scent, leading male moths on a futile quest and significantly hindering their ability to locate actual females. Consequently, fewer females are fertilized, which results in lower fecundity.

To incorporate the effect of mating disruption, we need to consider a local birth function, $b(n, A)$, that integrates this effect. To derive the form of $b$, we first derive an expression for the probability that a given female is fertilized (i.e., found by at least one male), $P(n, A)$, which depends on the population density, $n$, and the level of investment in mating disruption, $A$, in a given location. This probability can be calculated by assuming that males search for females independently from one another [26,29]. Then, the probability that a female is not found by any male is given by the Poisson coefficient $\exp(-\lambda n_m)$, where $\lambda_0$ is the probability that a given male finds a given female during the mating season, and $n_m$ is the number of males.

In turn, since the moth's sex ratio is approximately 1:1, we assume that $n_m = n/2$. We also assume, as in previous studies [29], that $\lambda$ declines with the investment in mating disruption and is given by $\lambda = \lambda_0/(1+aA)$, where $\lambda_0$ is the probability that a given male finds a given female without interference, and $a > 0$ is determined by the cost and the efficiency of the false pheromone flakes [29]. It follows that

$$P(n, A) = 1 - \exp\left(-\frac{\lambda_0 n}{2(1 + aA)}\right).$$

(6)

In turn, each fertilized female lays one egg mass, which implies that the number of egg masses, $n_0$, produced by the $n/2$ females, is given by

$$n_0 = \frac{nP(n, A)}{2}.$$

(7)

At low population densities, each egg mass develops, on average, into $r$ females ($2r$ adults). At higher densities, due to resource limitations at the larval stage, fewer eggs develop into adults.

To incorporate this, we consider the following birth function:

$$b = 2rn_0\left(1 - \frac{n_0}{k}\right). \tag{8}$$

The parameter values are estimated from the literature. Previous studies estimated that $r$ is between 2 and 10 [55,56], the carrying capacity is in the range of 1000 – 3000 egg masses per hectare [14], and $a$ is roughly 0.08 $USD^{-1}$ per hectare [26,29]. Studies have also shown that the annual dispersal distance of the spongy moth population, $\sigma$, ranges from 2.5 to 20 km [57] considering the dispersal kernel described in Eq (4). Lastly, to estimate $\lambda_0$ note that the dynamics described in Eqs (6–8) result in an Allee effect because, if $n$ is below a certain threshold, $n_a$, such that $b < 1$, the population declines. Empirical evidence shows that the Allee threshold is about 2–4 egg masses per hectare [58], namely, $n_a = 4r – 8r$ adults per hectare. The Allee threshold appears where $b = 1$, which implies that $\lambda_0 = (2/n_a)\ln(r/(r-1))$ [29]. For values of r between 2 and 10, this implies that $\lambda_0$ is between 0.17 and 0.0026. Note that some of these parameters could be scaled out as they could affect the units but not the underlying dynamics. Specifically, $\sigma$ only determines the units of $x$; $a$ only determines the units of $A$; and $n$ can be measured in units of the population's carrying capacity, in which case the single parameter $k\lambda_0$ would displace the parameters $k$ and $\lambda_0$.

## Cost of treatment, objective function, and constraints

Our goal is to find the treatment function $A(x, t)$ that minimizes the net cost of treatment over time while ensuring that $n$ propagates leftward at a rate that does not exceed $v$ (where $v$ is some target speed that is pre-determined and could itself be subject optimization, as described below). Specifically, $A$ also shifts leftward at a rate $v$, alongside the front, such that it has the form

$$A(x, t) = \hat{A}(x + vt), \tag{9}$$

where $\hat{A}(x)$ is stationary and does not depend on $t$. In turn, the annual cost of treatment (ACT) is defined as the cost of the treatment during one year over all spatial locations:

$$\text{ACT} = \int_{-\infty}^{\infty} \hat{A}(x)dx, \tag{10}$$

where the cost due to one unit of investment is normalized to one and could reflect the cost of expenditures plus the cost due to environmental side effects from the treatment. If $A$ is given by Eq (9), the ACT is the same in all years (because $\hat{A}$ does not depend on $t$), and $n$ approaches a quasi-equilibrium shape that only moves leftward along the $x$-axis at a rate $v$:

$$n(x, t) = \hat{n}(x + vt) \tag{11}$$

as $t \to \infty$, where $\hat{n}(x)$ has the shape of a population front. The objective is, therefore, to find a treatment function $\hat{A}(x)$ that minimizes the ACT under the constraint that $A(x, t) = \hat{A}(x + vt)$ would keep the front propagating at a rate $v$ (i.e., $n$ is given by Eq (11) as $t \to \infty$). We denote this optimal treatment, $\hat{A}(x)$, as $A^{\text{opt}}(x)$, and the population density $\hat{n}(x)$ associated with $A^{\text{opt}}(x)$ as $n^{\text{opt}}(x)$.

On top of that, note that the treatment and the population density must be non-negative (a negative treatment or population size would be biologically meaningless). Therefore, there are additional constraints, implying that

$$\hat{A}(x) \geq 0, \tag{12A}$$

$$\hat{n}(x) \geq 0, \tag{12B}$$

for all $x$. In particular, Eq (12B) implies that $\hat{A}(x)$ must not be too large, and the exact restrictions that Eq (12B) imposes on $\hat{A}(x)$ depend on the particular form of the dynamics.

## Novel algorithm for finding the optimal treatment strategy

Finding $A^{\text{opt}}(x)$ and $n^{\text{opt}}(x)$ for the dynamics given in Eqs (1, 3–5) (general model) or Eqs (2, 4, 6–8) (spongy moth model) is a complicated computational task. Here we developed a novel algorithm that solves this problem and finds either the optimal solution or some approximation to it. The general flow of the algorithm is described schematically in Fig 1: It begin with some population front, $\hat{n}$, and a corresponding treatment function, $\hat{A}$, such that the front propagates at the target speed $v(n(x, t) = \hat{n}(x + vt))$ if $A(x, t) = \hat{A}(x + vt)$. Then, the algorithm modifies the front's shape into one that necessitates a cheaper control function (with lower ACT) to still propagate at the same speed. The algorithm repeats this procedure until it can no longer improve $\hat{A}$ and $\hat{n}$ via local modifications of $\hat{n}$, i.e., it approaches a local optimum

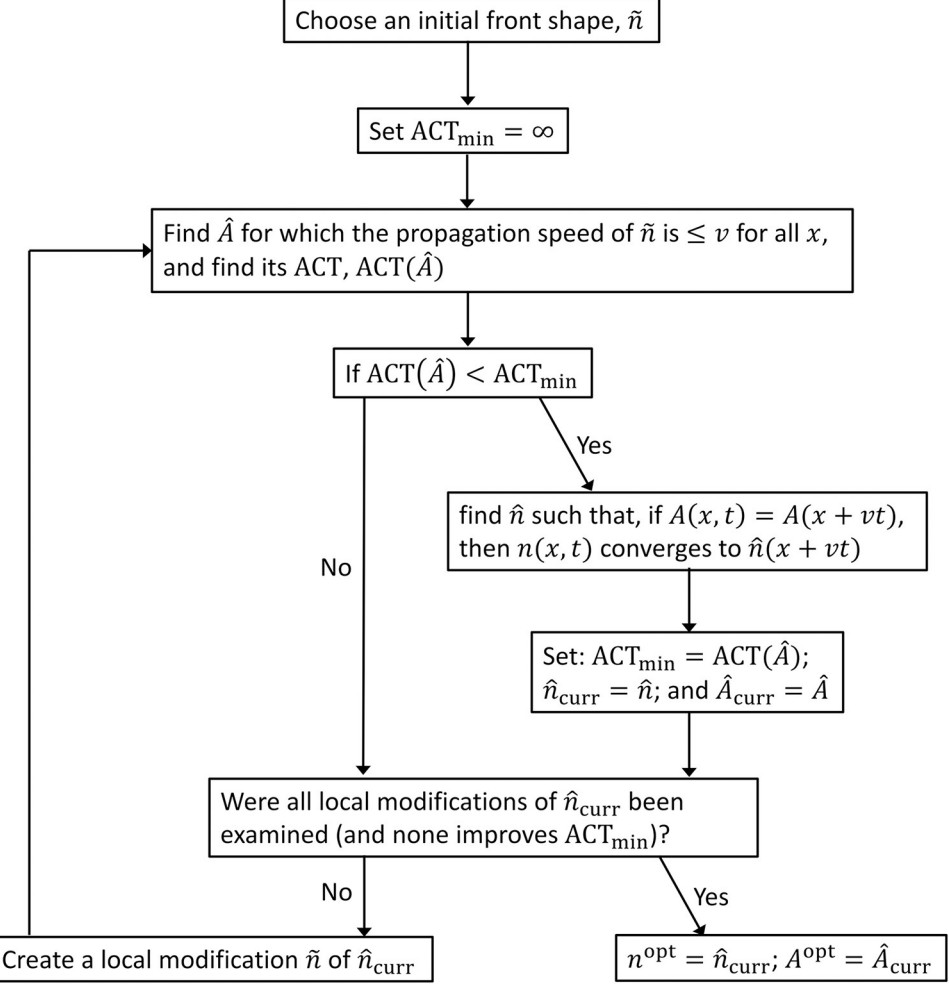

**Fig 1. Flow diagram summarizing the algorithm for finding the optimal treatment.**

in the space of all front shapes. To determine whether this local optimum is indeed the global optimum, the algorithm is applied to various different initial front shapes to verify that it always converges to the same solution. Although there is no guarantee that the algorithm will always converge to the global optimum, it at least provides an approximation to that optimum. Finally, since the algorithm gets $v$ as an input, we use it to explore the entire range of possible propagation speeds and find the optimal $v$. Note that this algorithm extends the algorithm developed in [21] to cases in which $v \neq 0$, and also to the study of more general response functions of the species to the treatment, such as those incorporated in Eqs (5) and (6–8). The complete code can be found on Zenodo [59]. Results as raw data can be found on Dryad [60].

## Algorithm: General framework

The algorithm initiates with some heuristic "guess" of a front, $\tilde{n}(x)$, which could be any function that is monotonically increasing with $x$, approaches 0 as $x \to -\infty$, and approaches the population's carrying capacity as $x \to \infty$. In turn, the algorithm implements a function "find $\hat{A}(\tilde{n})$" (see below) that calculates $\hat{A}$, the function that minimizes the ACT while guaranteeing that $\tilde{n}$ does not travel faster than $v$ in any location (e.g., if $A(x, t) = \hat{A}(x)$ and $n(x, 0) = \tilde{n}(x)$, then $dn(x, 0)/dt \leq v \cdot dn(x, 0)/dx$ for all $x$). The algorithm also implements a function "find $\hat{n}(\hat{A})$" (see below) that finds the front shape, $\hat{n}$, that is approached in the long run if $A(x, t) = \hat{A}(x + vt)$. (Note that $\hat{n}$ may differ from $\tilde{n}$ that is used to find $\hat{A}$ because there could be locations $x$ where $\tilde{n}(x)$ propagates slower than $v$.)

In the initial step, the algorithm uses these functions to calculate $\hat{A}$ associated with the initial front shape, and then to find $\hat{n}$ associated with that $\hat{A}$. Next, the algorithm examines several front shapes that are slight modifications of $\hat{n}$. Specifically, for numerical purposes, the algorithm iterates through all values of $x$ between $x_{\min}$ and $x_{\max}$ at some fine resolution $\Delta x$, where $x_{\min}$ is sufficiently small ($\hat{n}(x_{\min}) \approx 0$) and $x_{\max}$ is sufficiently large ($\hat{n}(x_{\max})$ is close to the population's carrying capacity). At each location $x_0$ within that range, the algorithm evaluates two modifications of $\hat{n}$: (a) $n^{\uparrow}(x_0)$, which differs from $\hat{n}(x)$ only at $x = x_0$ and its value there is between $\hat{n}(x_0)$ and $\hat{n}(x_0 + \Delta x)$, and (b) $n^{\downarrow}(x_0)$, which also differs from $\hat{n}$ only at $x = x_0$, where its value is between $\hat{n}(x_0)$ and $\hat{n}(x_0 - \Delta x)$. Specifically, $n^{\uparrow}(x_0) = [p\hat{n}(x_0) + \hat{n}(x_0 + \Delta x)]/(p + 1)$, where $p = 1,2,3\ldots$ defines the search's resolution.

In turn, for each modification, the algorithm calculates the associated treatment function $\hat{A}$ using "find $\hat{A}(\tilde{n})$", where $\tilde{n}$ is the examined modifications. If the modification results in an improvement ($\hat{A}$ associated with $n^{\uparrow}(x_0)$ or $n^{\downarrow}(x_0)$ for some $x_0$ results in a lower ACT than that associated with the original $\hat{n}$), then the algorithm calculates $\hat{n}$ that is associated with that modification (using "find $\hat{n}(\hat{A})$") and continues with that new function as $\hat{n}$. The algorithm repeats that step and continues to modify $\hat{n}$ and $\hat{A}$, until it converges to a value $\hat{n}$ and $\hat{A}$ that cannot be improved upon with local modifications, meaning any further local changes to $\hat{n}$ would result in a larger ACT. Also, the algorithm begins with a resolution $p = 1$ and keeps increasing that resolution until it does not find any improvement, even with the finer resolutions. We denote these final $\hat{n}$ and $\hat{A}$ as $n^{\mathrm{opt}}$ and $A^{\mathrm{opt}}$, respectively.

## Algorithm: Finding $\hat{A}$ that keeps a given front shape $\tilde{n}$ from propagating faster than $v$ in all locations ("find $\hat{A}(\tilde{n})$")

***Slow the spread, $v > 0$.*** First, note that the input function, $\tilde{n}(x)$, can be any function with a shape of a "front": $\tilde{n}(x)$ is monotone increasing with $x$ ($d\tilde{n}/dx \geq 0$), approaches 0 as $x \to -\infty$, and approaches the population's carrying capacity as $x \to \infty$. Initially, the algorithm determines

the time and space intervals, $\Delta t$ and $\Delta x$, respectively. If the dynamics are time-discrete, then the time intervals are always one generation ($\Delta t = 1$), and accordingly, the algorithm determines $\Delta x$ such that $v = \Delta x/\Delta t$, i.e., $\Delta x = v$. If the dynamics are time-continuous, $\Delta t$ is determined as the time during which the front would propagate a given $\Delta x$ if it propagates at a rate $v$ ($\Delta t = \Delta x/v$).

In turn, the algorithm simulates the dynamics of $n$ for a period $\Delta t$ with $A(x) = 0$ for all $x$, where initially $n(x, 0) = \tilde{n}(x)$. We denote the resulting function as $n_1(x)$ (i.e., $n1(x) = n(x, \Delta t)$). The requirement that the speed cannot exceed $v$ implies that, if $n_1(x_1) > \tilde{n}(x_1 + \Delta x)$ at some location $x = x_1$, then treatment must be used to suppress the population at $x_1$ (i.e., $\hat{A}(x_1)$ must be $> 0$). In turn, the algorithm seeks to find $\hat{A}(x)$—the "minimal" investment needed to ensure that the speed does not exceed $v$ in any location. Therefore, if $n_1(x_1) \leq \tilde{n}(x_1 + \Delta x)$ for some $x = x_1$, then $\hat{A}(x_1) = 0$, but if $n_1(x_1) > \tilde{n}(x_1 + \Delta x)$, then $\hat{A}(x_1)$ is such that it brings back $n_1(x_1)$ exactly to $\tilde{n}(x_1 + \Delta x)$ within $\Delta t$ time units.

Specifically, if the dynamics are time-discrete (Eq (2)), $\hat{A}(x)$ is such that, despite the possibility of having $n_1(x) > \tilde{n}(x + \Delta x)$, the birth rate of a population with $n_1(x)$ individuals is the same as the natural birth rate of a population with $\tilde{n}(x + \Delta x)$ individuals:

$$b(n_1(x), \hat{A}(x)) = b(\tilde{n}(x + \Delta x), 0) \tag{13}$$

if $n_1(x) > \tilde{n}(x + \Delta x)$, and $\hat{A}(x) = 0$ otherwise. In other words, $\hat{A}(x)$ is such that it reduces the effective population size from $n_1(x)$ to $\tilde{n}(x + \Delta x)$. In particular, considering the case study of the spongy moth, where $b(n, A)$ is given by Eqs (6–8), Eq (13) implies that $\hat{A}(x)$ is given by the following formula:

$$\hat{A}(x) = -\frac{1}{a}\left(\frac{\lambda_0 n_1}{log\left(\frac{\tilde{n}(\exp(-\lambda_0 \tilde{n})-1)}{n_1} + 1\right)} + 1\right), \tag{14}$$

where we denote $n_1 = n_1(x)$ and $\tilde{n} = \tilde{n}(x + \Delta x)$ to simplify the notations.

In turn, if the dynamics are time-continuous (Eq (1)), the rate at which $n(x)$ declines due to the treatment is given by $R(n(x), A(x))$. Therefore, the condition that $\hat{A}(x)$ reduces $n(x)$ from $n_1(x)$ to $\tilde{n}(x + \Delta x)$ within $\Delta t$ time units can be written as:

$$\Delta t = \int_{\tilde{n}(x+\Delta x)}^{n_1(x)} R^{-1}(n, \hat{A}(x)) dn \tag{15}$$

if $n_1(x) > \tilde{n}(x + \Delta x)$, and $\hat{A}(x) = 0$ otherwise, where $R^{-1}$ is the inverse function of $R$ with respect to $n$ (i.e., $R(R^{-1}(n, A), A) = n$). In the general case, $\hat{A}(x)$ can be found numerically for any given $x$ from Eq (15) via the Newton–Raphson method [61] since $R(n, A)$ is monotone increasing with $A$. In some special cases, however, a closed-form solution to $\hat{A}$ exists, which could simplify the procedure. Particularly, if dynamics are time-continuous and $R(n, A) = \beta A n^\alpha$ (Eq (5)) with $0 \leq \alpha < 1$, then $R^{-1}(n, A) = (\beta A)^{-1} n^{-\alpha}$, and it follows from Eq (14) that

$$\hat{A} = \frac{1}{\beta \Delta t}\int_{\tilde{n}(x+\Delta x)}^{n_1(x)}\frac{dn}{n^\alpha} = \frac{1}{\beta \Delta t}n^{1-\alpha}\big|_{\tilde{n}(x+\Delta x)}^{n_1(x)} = \frac{n_1(x)^{1-\alpha} - \tilde{n}(x + \Delta x)^{1-\alpha}}{\beta(1 - \alpha)\Delta t}. \tag{16}$$

***Stop or reverse the spread, $v \leq 0$.*** Consider the case in which $v < 0$: The algorithm needs to find $\hat{A}$ that reverses the direction of the front's propagation (i.e., the invasive species is being back-propagated at a speed $-v$). In that case, within $\Delta t$ time units, $n_1(x)$ needs to decline within $\Delta t$ time units to become $\leq \tilde{n}(x - \Delta x)$, where $\Delta x$ is now defined as the distance traveled at a speed $-v$ within $\Delta t$ time units: $-v = \Delta x/\Delta t$. Therefore, if $v < 0$, the condition for $\hat{A}(x) > 0$ is

that $n_1(x) > \tilde{n}(x - \Delta x)$. Otherwise, everything else is the same here as in the algorithm for the case $v > 0$ except that $\tilde{n}(x - \Delta x)$ replaces $\tilde{n}(x + \Delta x)$ in Eqs (13–16). Finally, if $v = 0$, the condition for $\hat{A} > 0$ is that $n_1(x) > \tilde{n}(x)$, and accordingly, $\tilde{n}(x)$ replaces $\tilde{n}(x + \Delta x)$ in Eqs (13–16).

## Algorithm: Finding $\hat{n}$ from $\tilde{n}$ and $\hat{A}$ ("find $\hat{n}(\hat{A})$")

For a given $\hat{A}$, which has been calculated from a given front $\tilde{n}$ using "find $\hat{A}(\tilde{n})$," it is guaranteed that the propagation speed of $n(x, 0) = \tilde{n}(x)$ at $t = 0$ is $\leq v$ in all locations. (The speed equals $v$ where $\hat{A}(x) > 0$ but could be lower where $\hat{A}(x) = 0$). Namely, if we allow $n(x, 0) = \tilde{n}(x)$ to evolve according to Eq (1) with $A(x, t) = \hat{A}(x + vt)$, its shape may change because some regions of $n$ will move leftward at a speed $v$ while some regions will move slower.

We would like to find $\hat{n}(x)$ — the front shape that evolves in the long run and has a propagation speed of exactly $v$ if treatment is given by $\hat{A}$, i.e., $n(x, t) = \hat{n}(x + vt)$ as $t \to \infty$ if $n(x,t)$ evolves according to Eq (1) with $A(x, t) = \hat{A}(x + vt)$. To find $\hat{n}$, the algorithm begins with $n(x, 0) = \tilde{n}$ and simulates the dynamics of $n$ with $A = \hat{A}(x + vt)$, until $n$ converges to a front shape that moves at a rate $v$ but does not change its shape otherwise, i.e., $n(x, t) = \hat{n}(x + vt)$.

### Algorithm for non-stationary, discrete-time dynamics

According to the spongy moth population model described in Eqs (2, 6–8), large values of $r$ may lead to chaotic dynamics of the untreated population. Therefore, one cannot expect that the population density in the untreated area—where $x$ is large—will approach its carrying capacity. In such cases, the algorithm sets a location $x_1$ beyond which the population is not treated, and the natural dynamics of the population are simulated for $x > x_1$. This way, the algorithm still converges to some front shape and some treatment shape within the range $x < x_1$. In the present study, however, we restrict attention to cases where $r$ is sufficiently small and the population converges to its carrying capacity in the non-treated area, while future studies could use the algorithm to analyze a broader set of problems.

### Robustness check

The algorithm finds $n^{\text{opt}}$ by performing a local search in the space of all possible front shapes: it examines local modifications of $\hat{n}(x)$ until it reaches a configuration for which no local modification exhibits an improvement. Therefore, the solution $n^{\text{opt}}$ and its associated $A^{\text{opt}}$ correspond to a local minimum in the space of all front shapes. This means that it is possible that $n^{\text{opt}}$ and $A^{\text{opt}}$ are only approximations of the global optimum. To further explore whether there exists a better choice of $\hat{n}$ and $\hat{A}$, we repeat the algorithm with various different choices of the initial front shape $\tilde{n}$, and we verify that it converges to the same $n^{\text{opt}}$ and $A^{\text{opt}}$.

### Determining the optimal propagation speed

The algorithm we described here finds the optimal treatment, $A^{\text{opt}}(x)$, its ACT, and the corresponding long-term front shape, $n^{\text{opt}}(x)$, for a given value of $v$. By using this algorithm to find $A^{\text{opt}}$ and the ACT associated with it for various values of $v$, we can examine the interplay between the propagation speed and the cost of treatment, and find the optimal target speed $v$. Specifically, consider the net present value (NPV), given by minus the cost of treatment and the cost of damage over time:

$$\text{NPV} = -\int_0^\infty \int_{-\infty}^\infty [A(x, t) + C(n(x, t) - n(x, 0))]e^{-\delta t} dx \, dt, \tag{17}$$

where the constant $C$ is the annual cost per unit of an infested area due to the establishment of the invasive species there (we measure the infested area relative to the infested area at t = 0). Neglecting the short-term dynamics and assuming that $A(x,t) = \hat{A}(x + vt)$ and $n(x,t) = \hat{n}(x + vt)$, it follows that

$$\text{NPV} = -\int_0^\infty \text{ACT}^*(v)e^{-\delta t}dt - \int_0^\infty Cvte^{-\delta t}dt = -\frac{\text{ACT}^*(v)}{\delta} - Cv\int_0^\infty te^{-\delta t}dt = -\frac{\text{ACT}^*(v)}{\delta} - \frac{Cv}{\delta^2}, \quad (18)$$

where $\text{ACT}^*(v)$ is the ACT if $\hat{A} = A^{\text{opt}}$ for a given target speed $v$ (see a similar analysis in [44]).

In turn, we would like to find the value of $v$ that maximizes the NPV. Note that $-\infty < v \leq v_0$, where $v_0$ is the speed at which the population front propagates if $A = 0$. Therefore, the NPV is maximized in one of the following three cases: (a) no treatment, i.e., $A = 0$ and $v = v_0$ (optimal if the NPV always increases with $v$); (b) eradicate as fast as possible, i.e., $v \to -\infty$ (optimal if the NPV always decreases with $v$); or (c) control the species at some intermediate value of $v$ (where the NPV has a local maximum). Specifically, it follows from Eq (18) that

$$\frac{d\text{NPV}}{dv} = -\frac{1}{\delta}\left(\frac{d\text{ACT}^*(v)}{dv} + \frac{C}{\delta}\right). \quad (19)$$

Therefore, the optimal strategy is to abandon treatment ($A = 0$ and $v = v_0$) if $d\text{ACT}^*(v)/dv < C/\delta$ for all $v$ and eradicate as fast as possible if $d\text{ACT}^*(v)/dv > C/\delta$ for all $v$. Otherwise, the optimal value of $v$ is where

$$\frac{d\text{ACT}^*(v)}{dv} = -\frac{C}{\delta}, \quad (20)$$

and

$$\frac{d^2\text{NPV}}{dv^2} > 0 \Rightarrow \frac{d^2\text{ACT}^*(v)}{dv^2} < 0. \quad (21)$$

On the other hand, if $\text{ACT}^*(v)$ in convex for some $v = v_1$, it is more efficient to let the species propagate at a speed greater than $v_1$ for some time and then at a speed lower than $v_1$ for some time rather than letting it spread at a speed $v_1$ at all times. Note that Eq (20) has also been suggested also by [44], where here we also show how to calculate $\text{ACT}^*(v)$ from a more detailed population dynamical model.

## Results and discussion

### Optimizing the treatment improves its cost-efficiency significantly

The algorithm finds the optimal treatment, at least approximately, for a wide variety of birth and death functions, as well as for various functional forms of the invasive species' response to the treatment. Our results for several cases of interest show that optimizing the spatial distribution of the treatment may lead to significant improvement in cost-efficiency. Specifically, the optimal treatment may be significantly more cost-effective than a treatment designed to stop the front while keeping its original shape or keeping a shape with a gradual linear decline. In particular, slowing the front but keeping its shape similar to that of an untreated population would result in an annual cost of treatment (ACT) higher by tens or hundreds of percentages than the ACT of the optimal treatment that slows the population's spread to the same speed. This result appears in the analysis of the general model (compare Fig 2A where ACT = 28.58 and Fig 2D where ACT = 8.58) as well as in the analysis of the spongy moth case study

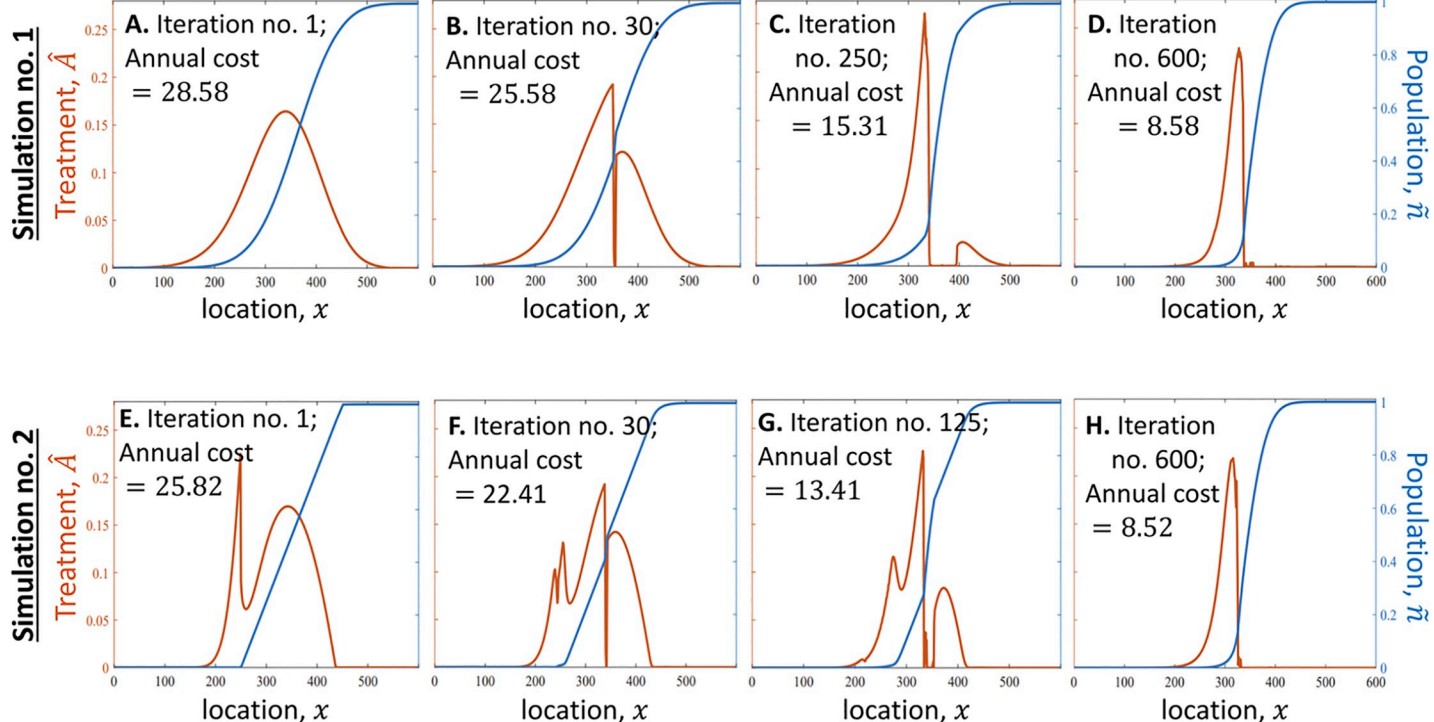

**Fig 2.** The algorithm finds the optimal treatment, which slows the population's propagation to a given target speed, $v$, while minimizing the annual cost of treatment (ACT) (simple model). **(A)** The algorithm begins with a population front, $\tilde{n}$, that has evolved naturally when the species has propagated at a speed $v_0$ ($v_0 > v$; blue line). The algorithm then finds the treatment function $\hat{A}$ for which the speed of the leftward movement of the front does not exceed $v$ in any location. **(B-D)** The algorithm finds new front shapes, as well as the treatment functions that hold these fronts propagating leftward at a speed $v$. In each iteration, the algorithm changes $\tilde{n}$ and $\hat{A}$ to those for which the ACT is lower. **(E-H)** The same algorithm as in (A-D) is executed, only here it begins with a piecewise-linear population front (E). In both simulation no. 1 (A-D) and no. 2 (E-H) of the algorithm, $\tilde{n}$ and $\hat{A}$ converge to a similar shape ((D) is similar to (H)), and we denote $\tilde{n}$ and $\hat{A}$ of this final outcome of the algorithm (shown in (D) and (H)) as $n^{opt}$ and $A^{opt}$, respectively. Parameters are the same in all panels: The target speed is $v = 10$ km/year, and the dynamics of $n$ follow Eq (1), where $b$ and $d$ are given by Eq (3) with $r = 2$ year$^{-1}$, $k = 2$, and $\gamma = 1$ year$^{-1}$; $G$ is given by Eq (4) with $\sigma = 1$ km; and $R$ given by Eq (5) with $\beta$ 1.25 USD$^{-1}$ year$^{-1}$ and $\alpha = 0.2$.

(compare Fig 3A in which ACT = $278.2K$ and Fig 3D in which ACT = $56.29$$, where ACT is measured in units of thousands of USD per one-kilometer strip of land over which the front propagates). A similar improvement in cost-efficiency is obtained if one uses the optimal treatment instead of a treatment that maintains a front with a shape of a straight line. This efficiency gain is evident in both the general model (compare Fig 2E where ACT = 25.82 and Fig 2H where ACT = 8.52) and the spongy moth case study (compare Fig 3A where ACT = $197.3K$ and Fig 3D where ACT = $56.15K$).

## The algorithm is robust

Our robustness check reveals that the algorithm is robust: it provides consistent results that are not sensitive to changes in initial conditions. Specifically, the algorithm converges to the same $n^{opt}$ and $A^{opt}$ for various choices of the initial front shape (Figs 2 and 3). This implies that the algorithm is robust and that it converges to either the optimum or to some approximation of the optimum that would be difficult to improve further. In particular, for the general model (Eqs 1, 3–5), the algorithm begins in one simulation with a front shape that emerges when no treatment is applied (Fig 2A), and in another simulation with a piecewise-linear front shape (Fig 2E), and it resulted in the same $n^{opt}$ and $A^{opt}$ in both cases (Fig 2D and 2H). We performed the same analysis with the case study of the spongy moth (Eqs 2, 4, 6–8; Fig 3), and

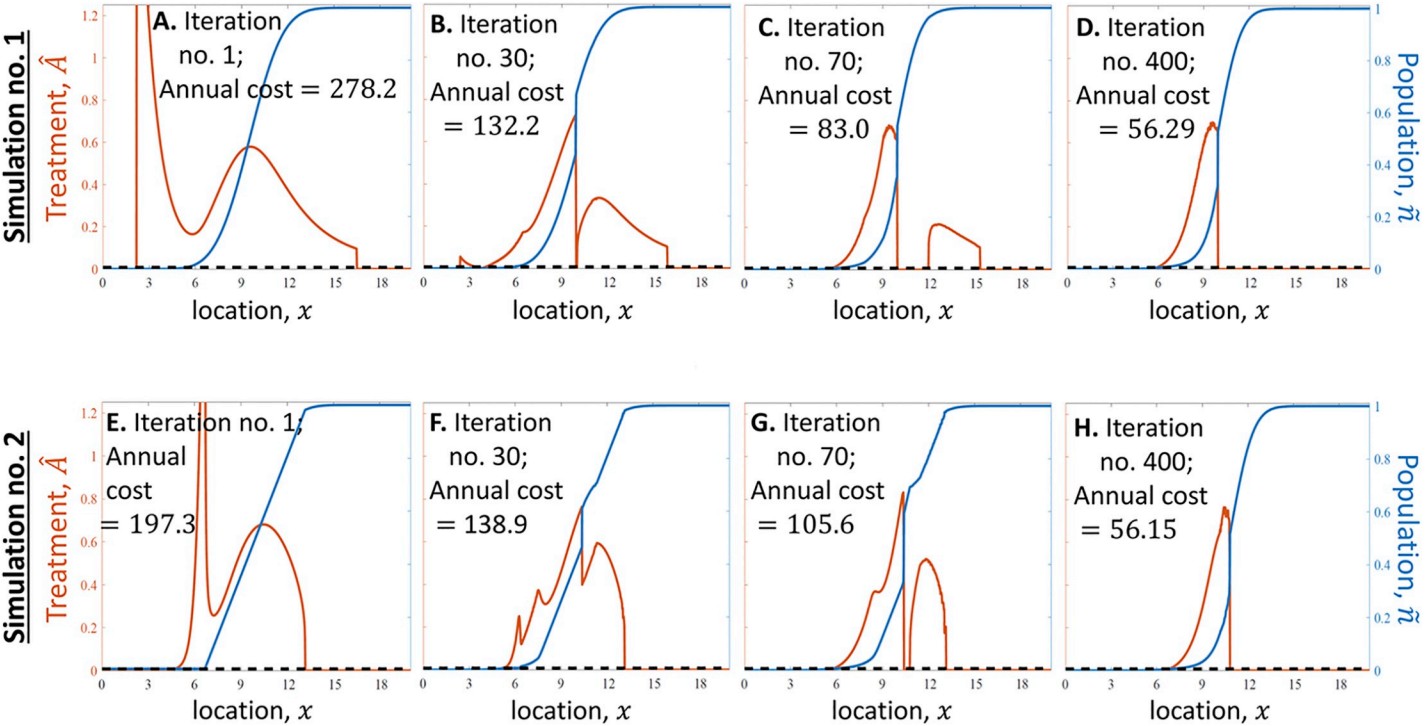

**Fig 3.** The algorithm finds the optimal treatment, which slows the population's propagation to a given target speed, $v$, while minimizing the annual cost of treatment (ACT) (spongy moth population model). The description of the panels is similar to that in Fig 2. **(A-D)** The algorithm begins with a population front $\tilde{n}$ that has evolved naturally (A); it finds front shapes that could be slowed with lower annual costs (B-D), until reaching a population front $\tilde{n} = n^{opt}$ for which ACT is minimized. **(E-H)** The same algorithm as in (A-D) is executed, only here it begins with a piecewise-linear population front (E). As in Fig 2, in both simulations no. 1 (A-D) and no. 2 (E-H) of the algorithm, the population front and the corresponding treatment function converge to a similar shape ((D) is similar to (H)). *Units*: Distance ($x$) is shown in units of $\sigma = 10$ km: population size ($\tilde{n}$) is in units of its carrying capacity; treatment cost ($\hat{A}$) is in USD per hectare per year; and ACT is given in thousands of USD per one-kilometer strip of land. *Parameters* are the same in all panels: The target speed is $v = 220$ m/year, and the dynamics of $n$ follow Eq (2), where $b$ is given by Eqs (6–8) with $r = 2$, $k\lambda_0 = 100$, and $a = 0.08$ USD$^{-1}$.

also in that case, the algorithm is robust and resulted in the same $n^{opt}$ and $A^{opt}$ for both initial choices of the front shape.

### The optimal spatial distribution of the treatment largely depends on the population's responses to the treatment

The analysis of the general model (Eqs (1, 3–5)) shows that, if the marginal cost per removal of one unit of $n$ is independent of n ($\alpha = 0$), the optimal strategy is to treat only those areas where the treatment reduces the population density to zero ($A(x) > 0$ only if $n(x) = 0$; Fig 4 –bottom row). However, if treatment becomes less efficient as $n$ decreases, as occurs in the general model (Eq (5)) with $0 < \alpha \leq 1$, the optimal treatment dictates treating in certain areas where $n(x) > 0$ (Figs 2 and 4 –top row). This is because, if $\alpha > 0$, it is more cost-effective to reduce the population in locations where its density is higher, and treatment in areas where $n(x)$ is larger reduces propagule pressure in areas where $n(x)$ is close to zero. This result is consistent with the result of [21], who examined the special case in which $v = 0$ and considered a similar (though not identical) model. They showed that, if $\alpha = 0$, treatment is needed only in locations $x$ where $n(x) = 0$ (i.e., only a "barrier zone" is needed); but if treatment efficiency is proportional to the population size ($\alpha = 1$), treatment is needed in some areas where $n(x) > 0$ (i.e., optimal treatment dictated treating in some "suppression zone"). Our result thus extends this result to cases where the manager

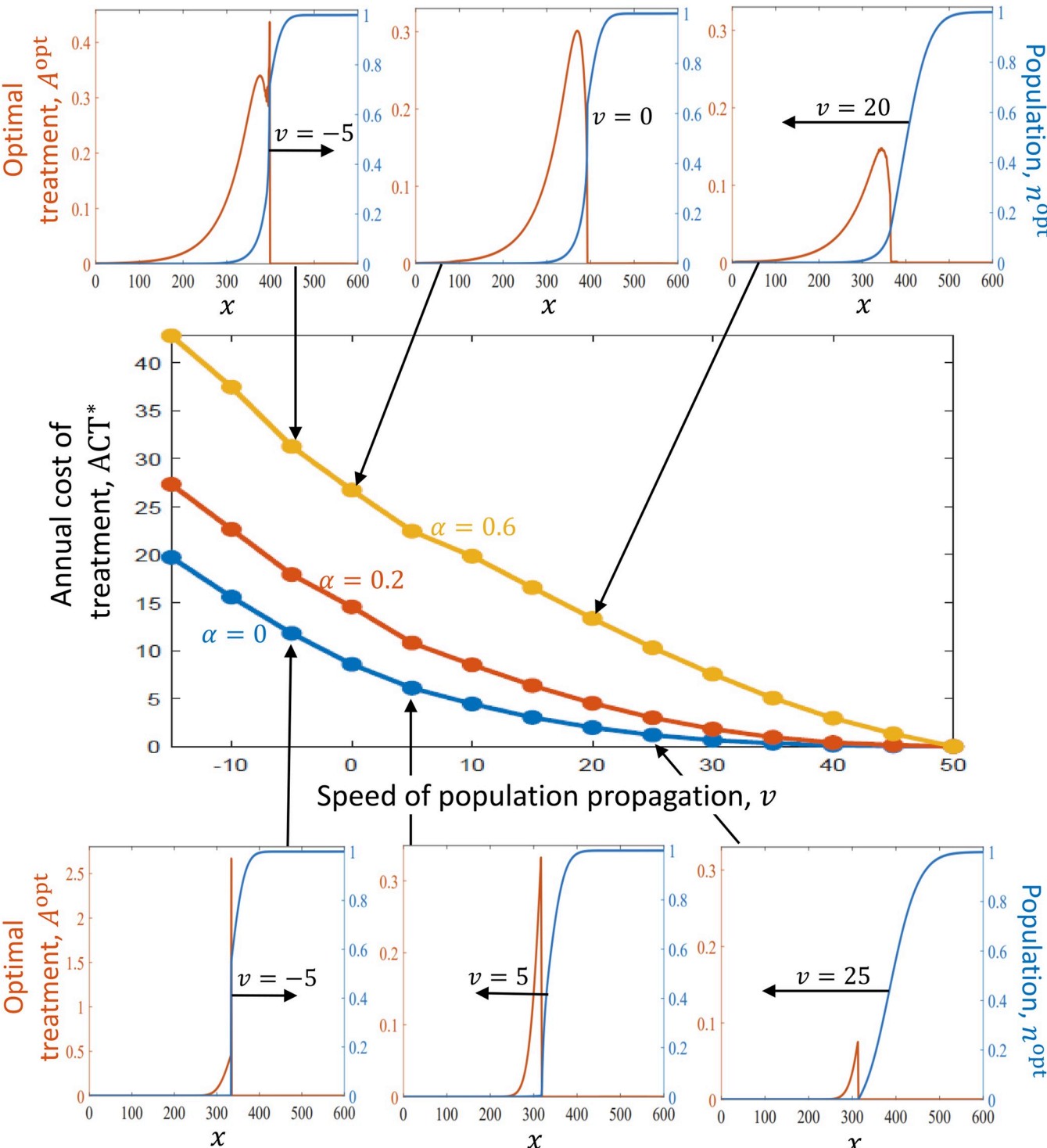

**Fig 4.** The annual cost of treatment decreases with the target speed $v$ and increases with $\alpha$ (simple model). The main (middle) panel shows the annual cost of treatment associated with the optimal treatment, ACT*, as a function of the target propagation speed of the population, $v$, for various choices of $\alpha$ (Eq (5)). ACT* decreases as $v$ increases, until $v$ equals the natural speed at which the species propagate without treatment $v = v_0 \approx 50$, where ACT* = 0. The ACT* is also higher when $\alpha$ is larger. The sub-panels demonstrate the shape of the optimal treatment profile, $A^{\mathrm{opt}}$, and the population density, $n^{\mathrm{opt}}$, as a function of the location, $x$. The three sub-panels on the top row show results for $\alpha = 0.6$, and those on the bottom row for $\alpha = 0$. Each sub-panel shows the optimal solution for a given target speed $v$, where both the treatment and the population density continuously move leftward at that speed (Eqs (9, 11)). The sub-panels demonstrate that, if $\alpha = 0$, treatment is applied only where $n(x) = 0$, whereas if $\alpha = 0.6$, treatment is distributed over broader areas, including some areas where $n(x) > 0$. If $v < 0$ (left sub-plots), a large concentration of treatment is peaked at the border between (a) the region where the population density is large and (b) the region

where the rest of the treatment is applied. All the parameters except $\alpha$ and $v$ are the same in all the panels: Eq (1) is considered, where $b$ and $d$ are given by Eq (3) with $r = 2$ year$^{-1}$, $k = 2$, and $\gamma = 1$ year$^{-1}$; $G$ is given by Eq (4) with $\sigma = 25$ km$^{-1}$; and $R$ given by Eq (5) with $\beta = 1/(1 - \alpha)$ USD$^{-1}$ year$^{-1}$. The raw data with the results of all the simulations for each $\alpha$ and $v$ can be found on Dryad [60].

slows the spread ($v > 0$) or reverses the spread ($v > 0$) and to cases where $0 < \alpha < 1$. In turn, Baker and Bode (2016) also examined optimal distribution of treatment for stopping the spread in a case where treatment efficiency is proportional to $n$. They found that, similar to our results, treatment should decline gradually in the area that needs to be protected from the species.

In turn, the analysis of the spongy moth population case study, in which treatment is implemented via mating disruption (Eqs (2, 4, 6–8)), reveals that the optimal strategy is to use mating disruption over a wide area that includes the regions where the population is below the Allee threshold but also a region where the population is above that threshold (Figs 3 and 5).

### To reverse the spread, the manager should apply the treatment within a narrow area

The spatial distribution of the optimal treatment also depends on the target propagation speed, $v$. If $v \geq 0$, our results show that $A^{\text{opt}}(x) > 0$ only if $x$ is smaller than a certain threshold, and $A$ varies smoothly within the range where it is positive (Figs 4 and 5). However, if $v < 0$, there is an apparent difference between the general model and the spongy moth model. Following the model of the spongy moth with mating disruption (Eqs (2, 4, 6–8)), $A^{\text{opt}}(x)$ remains a smooth function when $v$ is negative (Fig 5). But following the general model (Eqs (1, 3–5)), the optimal strategy when $v < 0$ is to treat very aggressively within a narrow area that creates a border between (a) the area where the species' density is low and (b) the area where the species is established and its density is close to its carrying capacity (Fig 4). No additional treatment is applied in the area where the species' density is high, and some treatment is applied in the area where its density is low in order to reduce it further to zero. This treatment profile moves to the right alongside with the population front at a speed $-v$ (Eqs (9, 11)). This implies that the optimal strategy for reversing the spread is to "push" the invasive species backward from the area where its density is low toward the area where its density is high.

### Tradeoff exists between the target propagation speed and the annual cost of treatment

The annual cost of treatment depends largely on the target propagation speed, $v$, and on the response of the species to the treatment. Our results reveal a tradeoff between slowing the spread and the annual cost of the optimal treatment, ACT* ($v$): In both the general model and the spongy moth model, slowing the spread more aggressively necessitates higher investments (i.e., lower $v$ results in a higher ACT*; Figs 4 and 5). This tradeoff has been suggested by Sharov & Liebhold (1988), who examined specific ACTs that were taken as given (without optimizing a front). In addition, population's response to the treatment also affects the ACT*. In the general model, the larger $\alpha$ is, the more expensive it is to slow the population's spread (Fig 4). In the gypsy moth population model, a larger $k\lambda_0$ implies that the Allee effect appears at lower population densities, which makes it more difficult to slow the spread. Accordingly, the larger $k\lambda_0$ is, the more expensive it is to slow the population's spread and the faster ATC increases as $v$ decreases (Fig 5). Moreover, in both the spongy moth and the general model with $0 \leq \alpha < 1$, stopping the spread ($v = 0$) and even reversing the spread ($v < 0$) can be achieved by investing a finite cost, and this may be optimal if $C/\delta$ is sufficiently large. In turn, note that in the general

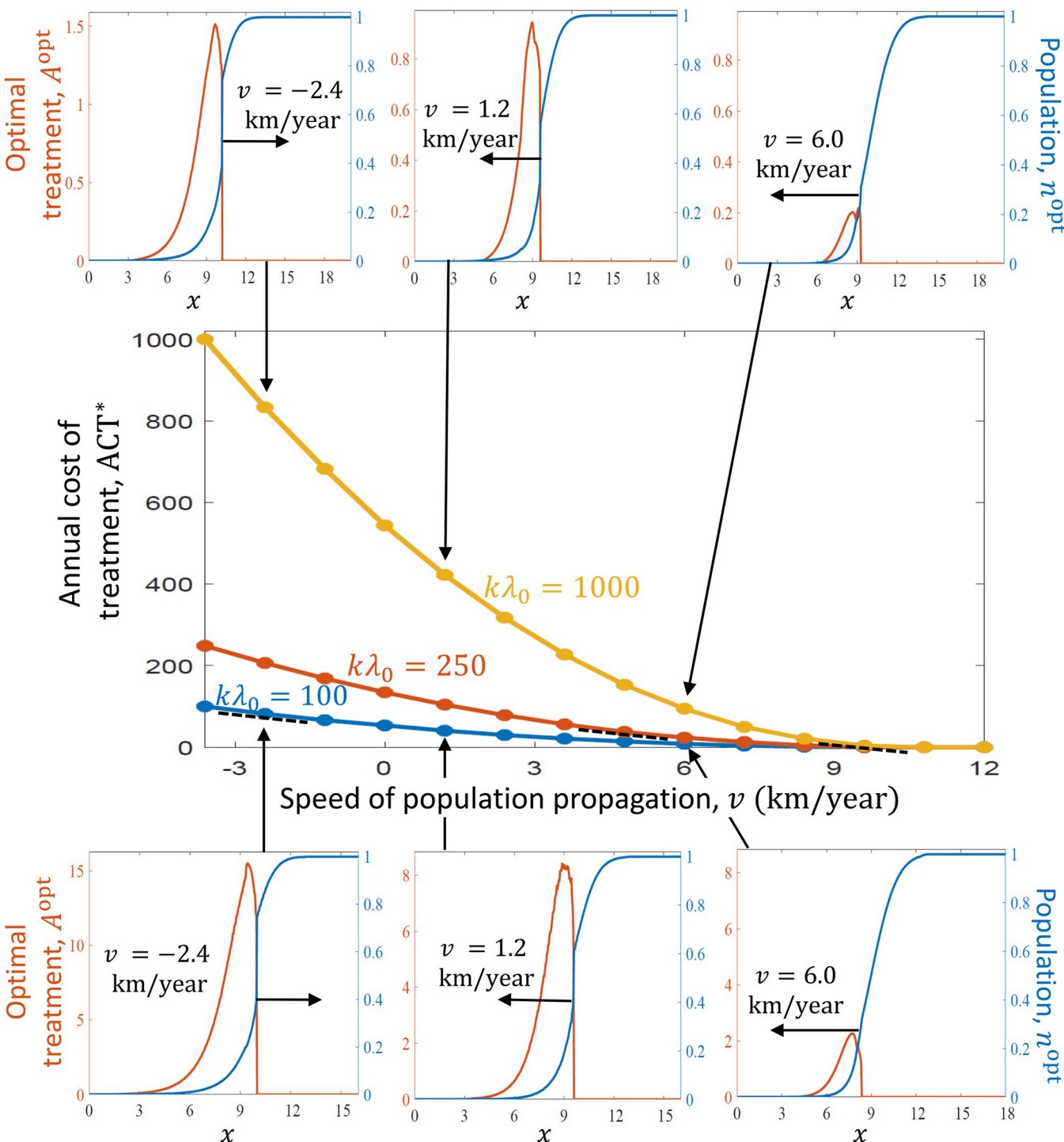

**Fig 5.** The annual cost of treatment decreases with the target speed $v$ and increases with $k\lambda_0$ (spongy moth population model). As in Fig 4, the main (middle) panel shows the annual cost of treatment associated with the optimal treatment, ACT*, as a function of the target propagation speed $v$ for three choices of $k\lambda_0$. ACT* decreases as $v$ increases, and reaches zero when $v$ equals the species' natural propagation speed without treatment ($v = v_0 \approx 12$ km year$^{-1}$). Higher $k\lambda_0$ values, indicating a lower Allee threshold relative to carrying capacity, resulting in an increased ACT*, where ACT* is measured in units of thousands of USD per one-kilometer strip of land. The dashed black lines show, for each line, where the marginal cost of slowing the spread (slope of ACT*) equals the marginal benefit of slowing the spread by one kilometer over a one-kilometer strip, estimated as 16K USD per year by [62]. In particular, the optimal speed according to this estimation is $v$ = -2.4 km/year if $k\lambda_0$ = 100; $v$ = 4.8 km/year if $k\lambda_0$ = 300; and $v$ = 9.6 km/year if $k\lambda_0$ = 1000. In turn, the sub-panels demonstrate the optimal treatment profile ($A^{opt}$), and population density ($n^{opt}$) across locations ($x$ in units of $\sigma$ = 10 km). Each sub-panel shows the optimal solution for a specific target speed $v$, with both treatment and population density advancing leftward at this speed (Eqs (9, 11)). All the parameters except $\alpha$ and $v$ are the same in all the

panels: Eq (2) is considered, where $b$ is given by Eqs (6–8) with $r = 2$ and $a = 0.08$ USD$^{-1}$, and $G$ is given by Eq (4). The raw data with the results of all the simulations for each $k\lambda_0$ and $v$ can be found Dryad [60].

model with $\alpha = 1$, it becomes impossible to stop the spread because ACT* approaches infinity as $\alpha$ approaches one from below.

## The optimal target propagation speed increases as the cost of damage decreases

The optimal choice of $v$, which maximizes the NPV (Eq (18)), can be found from the function ACT* ($v$) (Fig 4) together with the ratio between the cost of damage due to the species per unit area, $C$, and the discount rate, $\delta$ (Eq (20)). In particular, if $C/\delta$ is below a certain threshold, such that the slope of ACT* ($v$) is greater than $C/\delta$ even near $v = v_0$, then abandoning the treatment ($A = 0$ and $v = v_0$) is optimal. But if $C/\delta$ is above that threshold, the optimal choice of $v$ is where ACT* ($v$) is concave and its slope equals $C/\delta$ (Eq (20)) [44]. In turn, Figs 4 and 5 demonstrate that ACT* ($v$) (Eq (21)) is a concave function for a wide range of treatment methods and propagation speeds, which implies that the marginal cost of slowing the spread by additional kilometer per year increases as v decreases. Therefore, the optimal choice of $v$ is smaller if $C$ is larger.

In particular, [62] have conducted a thorough study to valuate the damage incurred by the spongy moth. They found that slowing the spread by one kilometer annually across a one-kilometer strip yields a benefit of 16K USD per year. (Specifically, Leuschner et al. (1996) found that the benefit of slowing the spread by 4 kilometers over a strip of about 800 Kilometers in length is worth 795M USD total or 50.9M USD per year, and therefore, the annual benefit per kilometer square is 50.9M/(4 800) = 16 K USD.) Our results show that, for parameter values used in Fig 5, the optimal target speed is $v = -2.4$ km/year if $k\lambda_0 = 100$; v = 4.8 km/year if $k\lambda_0 = 250$; and $v = 9.6$ km/year if $k\lambda_0 = 1000$. These results, particularly those where $k\lambda_0 = 100$ or $k\lambda_0 = 300$, are in line with previous conclusions [63]. These results imply that the strong Allee effect is a major reason why it could be cost-effective to slow the spread of the spongy moth.

## Limitations & future directions

Our study has several limitations, which could be addressed in future studies. First, the algorithm proposed here is tailored to optimizing the containment of an invasive species, which is only one component in a more comprehensive control program that could address prevention surveillance [64]. Second, the algorithm finds the spatial distribution of the population and the treatment that optimizes the annual costs in the long run. Additional research is needed to determine the most cost-effective management strategies during the initial phase—until the population front reaches its asymptotic shape. Third, the algorithm searches for the optimal front shape locally, and therefore, there is no guarantee that it always finds the global optimum. Fourth, the model does not incorporate the construction of physical barriers that prevent dispersal, such as fences and dams that restrict the movement of mammals [65] and aquatic species [18], and future research is needed to examine how to incorporate such strategies [36]. Fifth, we restricted attention to non-chaotic parameter regimes of the discrete-time dynamics. The algorithm can work in the chaotic regimes, but future studies are needed to analyze these cases. Finally, the dynamics in our model are deterministic, whereas real-world invasions exhibit stochastic dynamics and uncertainty. In particular, invasion could appear in more distant location and the locations where new populations appear might not be predictable. Future studies could

address this issue by examining how to combine suppression with surveillance that exposes the distribution of the invader and detects areas to which it spreads faster [39,66].

## Author Contributions

**Conceptualization:** Adam Lampert.

**Formal analysis:** Adam Lampert.

**Investigation:** Adam Lampert.

**Methodology:** Adam Lampert.

**Validation:** Adam Lampert.

**Writing – original draft:** Adam Lampert.

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
