## [Decision Letter · Decision Letter 0]

3 Jan 2024

Dear Prof. Lampert,

Thank you very much for submitting your manuscript "Optimizing strategies for slowing the spread of invasive species" for consideration at PLOS Computational Biology.

As with all papers reviewed by the journal, your manuscript was reviewed by members of the editorial board and by several independent reviewers. In light of the reviews (below this email), we would like to invite the resubmission of a significantly-revised version that takes into account the reviewers' comments.

In particular, both reviewers emphasize the importance of showing the application of the model/algorithm to a real-world case study.

One small typo: on line 192 it should be "heuristic" instead of "hyuristic"

We cannot make any decision about publication until we have seen the revised manuscript and your response to the reviewers' comments. Your revised manuscript is also likely to be sent to reviewers for further evaluation.

Sincerely,

Roger Dimitri Kouyos

Academic Editor

PLOS Computational Biology

Zhaolei Zhang

Section Editor

PLOS Computational Biology

Reviewer's Responses to Questions

**Comments to the Authors:**

Reviewer #1: This paper presents an algorithm that is capable of identifying the efficient control of an invader in an general spatial-dynamic model. The species spreads via a dispersal kernal. The marginal efficiency of treatment depends on the adult population density which varies over space. As such the efficient solution is not a uniform level of control at all points in space. Costs savings can be generated by applying varying levels of control at different points in space. The optimal control identifies where and when to apply control to achieve a target rate of propagation.

I enjoyed reading this paper. It is a nice example of using computational tools to address a growing biological problem. I also don’t see anything fundamentally wrong with the algorithm itself. However, I do have three general concerns about the way the model results are framed and presented to readers followed by several specific question/suggestions.

1. I would like to see the model applied to a real-world case study. While the author does apply the algorithm to a generic example, I think an application to a specific invasive species will be beneficial for two reasons. First, it will provide an indication of the magnitude of cost savings achieved with the algorithm. Yes there are theoretical cost savings that can be achieved with the algorithm, but it would help to indicate to the reader how large these cost savings may be. This is exactly the type of exercise that an algorithmic approach to invasive species control should provide over dynamic programming of Pontryagin’s maximum principle. The second reason for the case study is it will give the reader an indication of the data required to believably apply the algorithm. Reliable data on the density of invasive species at different points in space is notoriously difficult to obtain. The author should convince the reader that the data needed to benefit from the algorithm exists.

2. More effort needs to be devoted to convincing readers that the algorithm provides a good approximation of the actual solution. One approach would be to demonstrate how well the algorithm approximates the closed form solution presented in the paper. It is important to present both the global error as well as showing how the error varies over space. My guess is that the approximation error gets worse near the population front due to the nonlinearities.

3. Can the model accommodate species that spread via human-mediated, long-distance dispersal where populations establish well in advance of the primary invasion front? This would seem to allow for more complex functions for n(x,t) that may give the algorithm trouble. Or are you assuming that delta_t is long enough to eliminate any concerns about isolated populations?

Reviewer #2: This is a well-crafted and useful paper on optimal containment of a biosecurity event. I would only encourage more care with the existing literature so that readers can properly judge the value of the paper. For example:

1. "Few bioeconomic studies have explored the optimal containment of an invasive species

to stop or slow its spread." See:

https://www.sciencedirect.com/science/article/pii/S0921800911003855

and

https://onlinelibrary.wiley.com/doi/abs/10.1111/1467-8489.12305

and

https://esajournals.onlinelibrary.wiley.com/doi/abs/10.1002/eap.2319

to name a few!

2. "One reason for the scarcity of studies that explored optimal approaches to slowing61

populations’ spread is that standard optimization algorithms are limited in their capacity to62

solve spatially-extended dynamical optimization problems over large areas." Check, for example:

https://esajournals.onlinelibrary.wiley.com/doi/abs/10.1002/eap.2449

Otherwise the method is valuable and worth publishing in some form. The algorithm for finding an optimal treatment strategy is certainly a nice innovation, with clear results.

3. Also, I would only hope for some additional and *practical* application of the method and algorithm -- a case study that can illustrate the framework and its practical application. It is a big ask, but I think necessary and I can think of a number of potential case studies already (e.g., the attempt to eradicate and contain RIFA in Australia; the eradication of Papaya Fruit Fly in Queensland, many in North America, etc).

**Have the authors made all data and (if applicable) computational code underlying the findings in their manuscript fully available?**

Reviewer #1: Yes

Reviewer #2: Yes

PLOS authors have the option to publish the peer review history of their article (what does this mean?). If published, this will include your full peer review and any attached files.

Reviewer #1: No

Reviewer #2: No
---

## [Decision Letter · Decision Letter 1]

13 Mar 2024

Dear Prof. Lampert,

We are pleased to inform you that your manuscript 'Optimizing strategies for slowing the spread of invasive species' has been provisionally accepted for publication in PLOS Computational Biology.

Best regards,

Roger Dimitri Kouyos

Academic Editor

PLOS Computational Biology

Zhaolei Zhang

Section Editor

PLOS Computational Biology

Reviewer's Responses to Questions

**Comments to the Authors:**

Reviewer #1: I am now satisfied with the manuscript. The inclusion of a real world, practical application greatly improves the paper. I recommend that the paper be accepted for publication.

**Have the authors made all data and (if applicable) computational code underlying the findings in their manuscript fully available?**

Reviewer #1: None

PLOS authors have the option to publish the peer review history of their article (what does this mean?). If published, this will include your full peer review and any attached files.

Reviewer #1: No

---

## [Editor Report · Acceptance letter]

27 Mar 2024

PCOMPBIOL-D-23-01728R1 

Optimizing strategies for slowing the spread of invasive species

Dear Dr Lampert,

I am pleased to inform you that your manuscript has been formally accepted for publication in PLOS Computational Biology. Your manuscript is now with our production department and you will be notified of the publication date in due course.

With kind regards,

Anita Estes
